# A Comparison of Faculty and Student Acceptance Behavior toward Learning Management Systems

**DOI:** 10.3390/ijerph18168570

**Published:** 2021-08-13

**Authors:** Jinkyung Jenny Kim, Yeohyun Yoon, Eun-Jung Kim

**Affiliations:** School of Hotel and Tourism Management, Youngsan University, Busan 48015, Korea; jennykim1120@gmail.com (J.J.K.); yeohyun.yoon@gmail.com (Y.Y.)

**Keywords:** learning management system, faculty, student, higher education, technology acceptance model

## Abstract

During the COVID-19 pandemic, learning management systems have become the primary channel for lecturing and learning in higher education contexts. The present study investigates the development of user acceptance behavior toward a learning management system through use of the extended technology acceptance model. Moreover, this research identified differences between faculty and student behavior in a university environment. Based on a quantitative approach, the analysis results revealed that the main triggers of user acceptance behavior are self-efficacy, enjoyment, and computer anxiety. This study also documented the different influencing factors between faculty members and student groups, respectively. This work is expected to add to existing knowledge and help guide those working in higher education settings to establish more effective strategies for the optimization of learning management systems.

## 1. Introduction

The coronavirus disease (COVID-19) crisis caused radical changes in the tertiary education landscape; social distancing measures for those who have to be on campus and redesigned curricula for full online delivery were the two main areas of sudden change highlighted by higher education institutions across 20 countries [1]. Various online tools were utilized to try and maintain the same sense of depth of interaction between faculty and students as in face-to-face learning, and most universities made heavy use of learning management systems (LMS), which is online software for academic lecturing and learning, as a solution to manage educational programs [1,2]. Teaching staff had to scramble to set up and deliver remote lecturing and course materials through their institution’s LMS; at the same time, students were forced to switch to online systems for a new way of learning.

A number of studies on the technology acceptance model (TAM) have confirmed its predictive power in technology adoption behavior in individuals from various sectors, including education [3,4]. TAM holds that two essential determinants (i.e., perceived ease of use and perceived usefulness) of a user’s attitude, which are an individual’s favorable or unfavorable evaluation toward a particular technology, induce behavioral intentions, which show the likelihood of a person’s acceptance of a technology and subsequent actual use [5]. Despite the wide application of TAM, some have criticized that it is too limited to predict an individual’s technology adoption behavior. For example, Bagozzi [6] claimed that TAM involving only two indicators is not sufficient to predict user behavior toward a variety of technologies across different contexts. Hence, there has been a drive to incorporate other fundamental factors into the original TAM to better predict technology acceptance in higher education [1,3,7]. Among the incorporations or extensions to TAM, the most popular adopted external factors are: self-efficacy, subjective norm, enjoyment, and computer anxiety [8,9].

A common feature in higher education since the outbreak of COVID-19 is the separation of instructors and students by space and/or time, with both groups using technology, such as in the form of a LMS or a particular online class platform, for teaching and learning processes. Even though LMS is not entirely new to faculty and students, the main usage of LMS prior to COVID-19 had not, by and large, been for the direct purpose of teaching and studying; instead, LMS had been treated as a form of supplemental tool to facilitate communication and the sharing of notices [10]. One outcome of the transition from offline classes to online classes during COVID-19, according to perception of both staff and students, is that the experience falls short of expectations [11]. Education professionals widely admit that online university classes pose a number of challenges with respect to network infrastructure and that insufficient time was had to make a quick switch, which resulted in underprepared online classrooms [2]. Additionally, in general, students are relatively young compared to the instructors, and they are often reckoned to be more familiar with digital systems [12]. Compared with a tech-savvy generation, some staff may experience more difficulty in managing lectures in online spaces [13,14]. To exacerbate the situation, there was a lack of advanced preparation, such as in how faculty could be supported in remote teaching and how learners should connect to, and effectively engage in, remotely delivered content. Furthermore, both instructors and students have had to adjust their teaching and learning styles almost overnight, even when it is recognized that online teaching and learning require their own skill sets that take time to properly acquire [15]. Hence, a comparative approach in examining the adoption of two primary users of a LMS, namely faculty and students, would contribute to a strategy for an effective inclusion of innovative technologies within higher education environments.

Existing studies pertaining to technology acceptance behavior in higher education are mostly posited from the standpoint of only the faculty [3,4] or are focused on just the student’s perspective [7]. There are only a handful of studies [16,17] that investigate the difference between faculty and students regarding the development of behavioral intent toward technology-powered services for lecturing/learning purposes. Meanwhile, in the wake of COVID-19, there is a large body of study that addresses future outlooks of education in conjunction with technical solutions [18,19]. However, there are insufficient empirical studies, specifically during the critical first year of COVID-19, when the use of a LMS as an online lecturing/learning tool became more or less mandated overnight. Therefore, the present research is motivated to fill the gaps in the literature through generating knowledge about the acceptance levels of faculty and students towards a LMS, respectively, and offering insights for a better adoption of a LMS. Likewise, this study was built on the extended TAM that incorporates the four external influencing factors of: self-efficacy, subjective norm, enjoyment, and computer anxiety. Beyond the responses to the current COVID-19 crisis, the efforts to understand the fundamental driving forces of intention to use a LMS as an online class tool and will pave the way for more flexible and open education systems in the long term.

## 2. Literature Review

### 2.1. Learning Management System (LMS) over Pre and Post the COVID-19 Outbreak

A LMS has been defined as “a self-contained webpage with embedded instructional tools that permit faculty to organize academic content and engage students in their learning” [20]. A LMS enhances educational processes because it can support interactive communication between instructors and students with no restrictions in terms of time and place. To improve the effectiveness of facilitating and learning environments, almost every university around the world has adopted a LMS as part of their teaching and learning tools [21]. However, LMS has, until recently, only been used for relatively limited purposes, such as for passing on announcements to students, posting course syllabi, and creating rudimentary discussion boards [10]. Similarly, the results of a survey conducted with around 17,000 faculty and 75,000 students indicated that the main use of a LMS was firstly for sharing contents, followed by interaction or engagement activities [21].

Following the outbreak of COVID-19, many things changed dramatically, with education being one of the most greatly affected areas. For example, the Centers for Disease Control and Prevention introduced recommendations for higher education professionals in the COVID-19 environment to reduce transmission risks and promoted engaging in remote lecturing/learning [22]. Many universities, accordingly, responded with a new curriculum designs that support virtual rather than face-to-face environments. From this, LMSs emerged as prime channels in higher education, and both staff and students faced significant changes [2]; however, delivering courses through the adoption of technology, such as LMS, did not succeed at all institutions. Some universities failed to run online classes due to a lack of resources and ended up ceasing tuition [23,24]. Online education delivered through a LMS has also often turned out not to deliver the same or similar levels of engagement between faculty and students [11]. As a specific example, both instructors and students have expressed discontent due to technical glitches [25]. Such complaints reveal that fully online offerings in higher education were hit by a lack of preparedness and appropriate training.

### 2.2. The Extended Technology Acceptance Model (TAM) in Higher Education

The extant literature recognizes TAM as having a high predictive power in explaining technology adoption behavior in the education context [2,3,4]. TAM suggests perceived ease of use and perceived usefulness as core determinants of an individual’s attitude toward accepting a technology, which in turn dictates their intentions of use and actual usage [5]. Perceived ease of use has been conceptualized as “the degree to which a person believes that using a particular technology would be free from effort” while perceived usefulness has been defined as “the degree to which a person believes that using a particular system would enhance his or her job performance” [5].

Even though TAM has long dominated in explaining the formation of individuals’ technology adoption behaviors in teaching and learning contexts [2,4], there have been increased calls for a systematic synthesis to establish an even clearer mechanism behind technology acceptance in higher education. Hence, there have been substantial efforts in incorporating various external factors (i.e., self-efficacy, subjective norm, enjoyment, and computer anxiety) into TAM to improve study results. Self-efficacy is defined as “an individual’s judgment of his or her capability to organize and execute the courses of action required to attain designated types of performances” [26]. Meanwhile, subjective norm is described as the degree of social pressure perceived as to whether to conduct a behavior or not [2]. Enjoyment refers to how users perceive activities and services to be enjoyable in a way that enriches their lecturing and learning experiences [8]. Finally, computer anxiety is explained as “the tendency of an individual to be uneasy, apprehensive, or fearful about the current or future use of computers in general” [27].

Bhattarai and Maharjan [28] investigated acceptance behavior of digital teaching-learning systems, and their results confirmed that perceived ease of use was mainly affected by social influence and enjoyment, whereas perceived usefulness was positively influenced by self-efficacy, social influence, and enjoyment. Likewise, Abdullah and Ward [8] carried out a quantitative meta-analysis of research that was published within the previous ten years to predict user technology adoption behavior in an e-learning context. Out of 107 papers, they identified self-efficacy, subjective norm, enjoyment, and computer anxiety as the most frequently examined external influencing factors of TAM. Similarly, Baki, Birgoren, and Aktepe [29] reviewed 203 studies based on TAM in online education, and found that self-efficacy, subjective norm, enjoyment, and computer anxiety had a significant impact on perceived ease of use and perceived usefulness. In short, self-efficacy, subjective norm, enjoyment, and computer anxiety have been individually and collectively adopted in the framework of TAM as external factors—that is, to make the extended TAM—to explain faculty and student adoption behavior around a LMS [14,30].

## 3. Hypotheses Development

### 3.1. Incorporating External Factors into TAM

A level of confidence, resources, and skills to use products and systems often illustrate a degree of self-efficacy and it has been adapted as an essential facilitator, which affects perceived ease of use and perceived usefulness in the education context [4,7]. For instance, the significant association between self-efficacy and perceived ease of use was discovered from instructors’ perspectives in the context of web-based learning systems [31]. Lee, Hsiao, and Purnomo [32] proposed computer self-efficacy and Internet self-efficacy as student characteristics, and their analysis results revealed the significant impact of Internet self-efficacy on perceived ease of use and perceived usefulness of an e-learning system. Fathema, Shannon, and Ross [3] incorporated self-efficacy in the original TAM to predict the acceptance of a LMS. Their results, which involved 298 responses from faculty members, indicated that self-efficacy influenced both perceived ease of use and perceived usefulness. That is, if individuals perceive themselves as having higher capabilities when it comes to using a LMS, they would find that the system is easy to use and useful. From this, we derived the following hypotheses:

**Hypothesis** **1a (H1a).**
*Self-efficacy significantly increases perceived ease of use.*


**Hypothesis** **1b (H1b).**
*Self-efficacy significantly increases perceived usefulness.*


Subjective norm was also classified as one of the salient influential determinants in the extended TAM [8,28]. Subjective norm has been expressed as being an individual’s perception that people who are meaningful to them think that they must or must not behave in a certain way [33]. Baleghi-Zadeh et al. [34] considered subjective norm to be an important factor in the extension of TAM to predict student intentions to use a LMS. Their results revealed the important role of subjective norm in enhancing perceived ease of use and perceived usefulness. Similarly, Fındık-Coşkunçay, Alkiş, and Ozkan-Yildirim [35] examined student adoption behaviors of a LMS, based on responses collected from 470 students, and they validated the strong association between subjective norm and perceived usefulness. Muhaimin et al. [36] centered on advanced Web 2.0 tools in improving lecturing and learning experiences and examined instructors’ intentions to use such technologies. They involved subjective norm through the framework of the extended TAM, and the analysis results validated that subjective norm exerted an influence on perceived ease of use and perceived usefulness. Hence, we posited the following hypotheses:

**Hypothesis** **2a (H2a).**
*Subjective norm significantly increases perceived ease of use.*


**Hypothesis** **2b (H2b).**
*Subjective norm significantly increases perceived usefulness.*


Meanwhile, numerous scholars have proposed enjoyment as a key factor determining the perceived ease of use and perceived usefulness when it comes to adopting an online education system [8,29]. Enjoyment refers to the degree to which individuals enjoy using a novel system which influences the perception of effort needed to use the system [30,37]. Thus, enjoyment has been commonly adopted as a significant facilitator in the extended TAM, and a number of studies have validated its positive influencing role on perceived ease of use and perceived usefulness in the context of online learning systems [28,38]. Al-Rahmi et al. [30] confirmed that student technology acceptance behaviors in higher education were mainly affected by perceived enjoyment, which in turn enhanced perceived ease of use and perceived usefulness. Similarly, hedonic motivation was discovered to play a fundamental role in generating academic adoption behaviors toward e-learning [39]. In particular, enjoyment has been frequently employed in mobile mediated learning systems [40]. Therefore, the following hypotheses were formulated:

**Hypothesis** **3a (H3a).**
*Enjoyment significantly increases perceived ease of use.*


**Hypothesis** **3b (H3b).**
*Enjoyment significantly increases perceived usefulness.*


Computer anxiety is another influential factor that has been frequently adopted in the extended TAM. Computer anxiety means exhibiting concern about using a computer [40], and individuals who possess a higher degree of anxiety around using computers are more likely to hesitate in LMS adoption. Linked to this notion, computer anxiety has been found to be a determining factor of perceived ease of use and perceived usefulness in the accepting of technology in higher education [8,9]. For instance, Ashtari and Eydgahi [41] focused on the increased usage of cloud computing technology in higher education and tested user acceptance of such technology. They conducted a survey at a U.S. university and showed that computer anxiety played a negative role in students’ perception of the usefulness of technology. Likewise, Baki and Birgoren [29] conducted a meta-analysis based on 203 studies on e-learning and noted that computer anxiety is a salient determinant of perceived ease of use and perceived usefulness toward e-learning systems. Based on these findings, we proposed the following hypotheses:

**Hypothesis** **4a (H4a).**
*Computer anxiety significantly decreases perceived ease of use.*


**Hypothesis** **4b (H4b).**
*Computer anxiety significantly decreases perceived usefulness.*


### 3.2. Relationships among Constructs Rooted in the TAM

Perceived ease of use and perceived usefulness are the core determinants that build individuals’ attitudes in TAM [5]. On the basis of numerous applications of TAM to higher education contexts, there is abundant evidence that supports the significant role of perceived ease of use and perceived usefulness in the development of attitudes [9,29]. In addition, the effect of perceived ease of use on perceived usefulness has been verified in many studies. For instance, Eraslan Yalcin and Kutlu [7] confirmed the significant impact of perceived ease of use on perceived usefulness in the acceptance of e-learning platforms at universities. At the same time, behavioral intentions in online learning contexts are described as being the level to which a user formulates conscious plans to use a LMS [41]. Many scholars have validated that attitude is one of the best indicators of intention to use new technology, which supports the TAM [3,8,38,42]. Similarly, Bhattarai and Maharjan [28] investigated student behavior regarding digital transformation in learning activities, and they provided empirical evidence of the strong link between attitude and intention to use technology. Consistent with the theoretical and empirical evidence in suggesting the appropriateness of TAM in the field of online education systems, the following hypotheses were derived.

**Hypothesis** **5a (H5a).**
*Perceived ease of use significantly increases perceived usefulness.*


**Hypothesis** **5b (H5b).**
*Perceived ease of use significantly increases attitude.*


**Hypothesis** **6 (H6).**
*Perceived usefulness significantly increases attitude.*


**Hypothesis** **7 (H7).**
*Attitude significantly increases intention to use.*


### 3.3. The Moderating Role of User Type

Today’s students are often perceived to function better and be more conversant with technology, which is why they should in theory be more suited to a digital learning environment than older teaching staff [12]. Students are therefore expected to be more motivated than faculty toward online education [43]. However, in actuality, not all students exhibit a high level of technology acceptance across different types of online educational platforms and tools. Moreover, Gong, Xu, and Yu [44] explained that service providers in educational settings, namely instructors, have substantial autonomy over their teaching methods, which include technology adoption and use, and consequently they have less peer competition for accepting novel technologies. At the same time, a LMS is often not used to facilitate the teaching process to its fullest capabilities by faculty members [3]. Faculty acceptance of a LMS is one of the barriers to widespread adoption of teaching in online spaces [14,45]. This implies the importance of identifying the triggers behind adopting a LMS as an online education channel and how the triggers differ depending on user type.

Existing studies do not provide sufficient evidence regarding the moderating role of user type in technology adoption behavior in higher education. According to Granić and Marangunić [9], 83 percent of research out of 71 relevant studies from 2003 to 2018 was subject to students’ adoption behavior, whereas only 6 percent consisted of technology acceptance from the viewpoint of faculty. Out of a relatively small amount of work that investigated the difference between faculty and students in using technology in their teaching and learning, some researchers have reported that there are no significant differences between these two groups [16,46]. On the other hand, several researchers have provided empirical evidence that indicates noticeable differences between faculty and students. For instance, Sampsel [17] examined how teaching staff and undergraduate students in nursing education perceive usefulness, effectiveness, and friendliness of remote-presence robotics differently. Their results revealed that the influence of such distance education technology on satisfaction varies according to different user type (i.e., faculty vs. student). Piotrowski and Pedagogical [47] examined faculty and student perspectives toward digital pedagogical tools and determined that faculty members were apparently disinterested in the online tools. In contrast, students displayed a positive attitude toward academic application of such tools. Smale, Regalado, and Amara [48] described through a case study at City University of New York how staff were overwhelmed by instructional technologies for teaching online, whereas students were frustrated due to the lack of 24/7 technical assistance for online learning. In light of the above, it can be said that the formation of acceptance behavior toward LMS differs between instructors and students in higher education, and the following hypotheses are suggested accordingly:

**Hypothesis** **8a–l (H8a–l).**
*The user type moderates the link among proposed study constructs.*


Accordingly, the proposed conceptual model is displayed in Figure 1.

## 4. Methodology

### 4.1. Sampling and Data Collection

In the COVID-19 pandemic situation, higher-education institutions are actively utilizing LMS to maintain student–faculty interactions, such as face-to-face classes and improve learning efficiency. A faculty delivers online lectures through a LMS, and students face the need to embrace this new online learning system. In order to test hypotheses, quantitative approach has been conducted based on survey data. A survey was conducted via Google Forms. The survey participants were students and faculty who have experience of using a LMS; the participants were all from universities in Busan, Korea’s second-largest city. Students and faculty who qualified and agreed to take a part in the survey were sent a link to the survey and then completed it by themselves. In order to increase respondent participation, a gift voucher was offered. A total of 400 surveys were collected over a period of four weeks. In order to analyze only the respondents who had LMS experience in higher education institutions suitable for this study among the collected answers, those who answered that they had no LMS experience in higher education institutions or those who did not indicate the length of their college years were excluded from the statistical analysis. 24 of the respondents (14 students and 10 faculty members) were excluded from the analysis. This resulted in a final data sample comprising 216 students and 160 faculty.

### 4.2. Measurements

All measurement items were rated according to a 7-point Likert-type scale, developed and tested in previous studies, anchored with “1 = strongly disagree” to “7 = strongly agree”. Self-efficacy, subjective norm, and enjoyment were measured using four items, and computer anxiety was measured using three items, as adapted from [7,40,41]. The theoretical framework of TAM explains that if a person perceives a certain technology to be easy to use and useful, it forms a positive attitude and influences intention to use this specific technology [5]. The measurement items for constructs rooted in the TAM—encompassing perceived ease of use and perceived usefulness—were assessed using four items, and attitude and behavioral intentions were evaluated using three items borrowed from [3,4,5].

### 4.3. Descriptive Information

For the student group and the faculty group, slightly more responses were received from men (56.9%/56.6%) versus women (43.1%/43.4%). For the online learning system experience, more than half of the student group (55.1%) reported having experience, while more than half of the faculty group (57.2%) had no experience. The majority of the student group consisted of 20–29-year-olds (87%), while the faculty group’s age range was more heavily weighted in the 40–49-year age band (38.1%). In the student group, freshmen (41.7%) accounted for the largest segment of respondents. In addition, the length of teaching career in the faculty group ranged from less than 5 years (20.1%) to more than 20 years (19.5%).

### 4.4. Measurement Model Analysis

This study employed partial least squares structural equation modeling (PLS-SEM), which is a comprehensive procedure used in multivariate statistical analysis to evaluate a proposed measurement model and structural model.

Following Henseler and Ringle [49], measurement invariance was tested using (a) configural invariance evaluation (step 1), (b) compositional invariance evaluation (step 2), and (c) assessment of the equal mean values and variances (step 3). In general, if partial measurement invariance is successfully established in the first two steps (step 1 and step 2) and the study focuses on cross-group comparisons with unpooled data [50], this can be deemed acceptable.

First, it is important to address the issue of configural invariance when discriminating whether different groups have the same construct and factor parameter coefficients [49]. The measurement model did not have any issues with convergent validity, because of the values of Cronbach’s α and composite reliability (≥0.7), AVE (≥0.5) (see Appendix A, Table A1). Furthermore, Appendix A (Table A2 and Table A3) show that the discriminant validity and the Fornell–Larcker criterion and the HTMT ratio of correlations statistic (<0.9) were acceptable [51].

Second, the value of original composite scorea correlation (*c*) was not lower than the 5.0% quantile of the permutation procedure (*c_u_*), and therefore, compositional invariance [49] was established between the student and faculty groups (see Appendix A, Table A4). Following from this, we compared the path coefficients between two groups (students and faculty) to assess significant differences.

The common method bias was evaluated using the variance inflation factor (VIF) value proposed by Kock [52]. The VIF value is lower than 3.3 (VIF_Student group_ = 1.532 − 2.574; VIF_Faculty group_ = 1.597 − 2.773), confirming that there is no threat of common method bias.

### 4.5. Structural Model Assessment

We evaluated the structural model fit using the following criteria with SmartPLS 3.3.2 [53]. To start with, the multicollinearity among exogenous variables was evaluated using a VIF. As mentioned before, the values of VIFs are smaller than 3.3, indicating multicollinearity is unlikely to be problematic. In addition, the predictive power of the model is not a concern, because R^2^ is more than 10% [54]. Then, the predictive relevance of the endogenous latent variable was achieved, because the values of Stone–Gesser (Q^2^), namely the cross-validated redundancy Q^2^ values, were greater than 0. Finally, the model fit was considered acceptable, because the value of the standardized root mean square residual (SRMR) is lower than 0.10 [55,56]. The value of SRMR is 0.059 for student group respondents and 0.060 for faculty group respondents.

## 5. Results

### 5.1. Hypotheses Testing

The findings show that self-efficacy (*β* = 0.490, *p* < 0.01; *β* = 0.259, *p* < 0.01), enjoyment (*β* = 0.214, *p* < 0.01; *β* = 0.2066, *p* < 0.01), and computer anxiety (*β* = 0.275, *p* < 0.01; *β* = 0.458, *p* < 0.01) significantly influenced perceived ease of use in the student and faculty groups, respectively, thus supporting H1a, H3a, and H4a in both groups. However, subjective norm (*β* = 0.128, *p* < 0.05) significantly influenced perceived ease of use in the faculty group. Therefore, H2a was only supported in the faculty group. Unexpectedly, computer anxiety did not significantly influence perceived usefulness in both groups, thus not supporting H4b. However, subjective norm (*β* = 0.219, *p* < 0.01) significantly influenced perceived usefulness in faculty group. Therefore, H2b was only supported in the faculty group.

The findings show that, as expected, perceived ease of use significantly affects perceived usefulness (*β* = 0.277, *p* < 0.01; *β* = 0.320, *p* < 0.01) and attitude (*β* = 0.232, *p* < 0.01; *β* = 0.345, *p* < 0.01), perceived usefulness (*β* = 0.697, *p* < 0.01; *β* = 0.565, *p* < 0.01) influences attitude, and attitude (*β* = 0.846, *p* < 0.01; *β* = 0.863, *p* < 0.01) affects intention to use in both groups. Hence, H5a and H5b, H6, and H7 were supported in both the student and faculty groups. Table 1 displays the details of the testing of the hypotheses.

### 5.2. Comparison between Student and Faculty Groups

We compared estimates between the student and faculty groups to identify differences in the framework of self-efficacy/subjective norm/enjoyment/computer anxiety–perceived ease of use–perceived usefulness–attitude–intention to use from the perspective of the moderating role of user type studies using multi-group analysis (MGA) with a partial least squares multi-group analysis (PLS-MGA) procedure [54]. As displayed in Table 1, significant group differences were only found for the impact of self-efficacy use (estimate of difference = 0.232, *p*-values of difference = 0.011, *p* < 0.05) and computer anxiety use (estimate of difference = −0.184, *p*-values of difference = 0.067, *p* < 0.10) on perceived ease of use. Therefore, H8a and H8g were supported. However, user type does not moderate other relationships in the student and faculty groups. Thus, H8 was found to have a partial moderating effect.

## 6. Discussion

This study sought to verify the differences between user types in technology acceptance behavior in higher education based on the extended TAM that incorporated the four external factors of self-efficacy, subjective norm, enjoyment, and computer anxiety. We addressed the limitation of considering only one standpoint of faculty and students and provide evidence of the user type’s moderating role.

In case of students group, the results indicated that self-efficacy, enjoyment, and computer anxiety have a significant influence on perceived ease of use of a LMS. The variables that had a significant effect on perceived usefulness were self-efficacy and enjoyment. However, in faculty group, self-efficacy, subjective norm, enjoyment, and computer anxiety were found to have an impact on perceived ease of use of a LMS. In addition to self-efficacy, subjective norm, and enjoyment significantly affected perceived usefulness.

The comprehensive results of the analysis determined that self-efficacy and enjoyment were significant contributors to increasing perceived ease of use and perceived usefulness in both the faculty and student groups. This lends evidence to the fact that if users have strong confidence in their ability to use a LMS, or enjoy LMS teaching and learning activities/services, they will feel it is easy to use and that it is useful. In fact, it is widely accepted that it is easy for students and professors with high self-efficacy to use a LMS [3,4,28,32]. Consequently, these confident individuals use a LMS more than less-confident individuals [3]. This is supported by Bandura [26] in that high self-efficacy leads to an active learning process.

Although subjective norm influenced faculty members’ perceived ease of use and perceived usefulness, it did not reveal any significant effect among the student group. This result indicates that subjective norm is important to faculty, and this result agrees with the study by Muhaimin et al. [36]. In addition, subjective norms, which were significant determinants only for the faculty group, disprove the importance of social influence on them. In other words, they want to meet the needs, expectations, and preferences of members of society who are important or valuable to them. Thus, we can expand the acceptance of LMS by strengthening it. Meanwhile, from the student group, it also supports the findings of Salloum et al. [38], which assume that they cannot be confident that a LMS would be easy to use and useful even if a LMS is socially preferred according to peers. While these findings were not in line with the results of other prior studies [8,28] in which subjective norm was shown to have a positive impact on perceived ease of use and usefulness in the student group.

Computer anxiety exerts a significant influence on perceived ease of use in both the student and the faculty groups, but not on perceived usefulness in both groups. Previous studies have argued that computer anxiety is a determinant of perceived ease of use and usefulness [9,29], but results show that concerns about computer use, whether faculty or students, have no significant relation to usefulness. In other words, both faculty and student groups feel the ease of use when they do not feel scared, uncomfortable, or nervous when using the computer, but they do not consider that it is useful to improve performance and achievement, and increase the learning effect.

Findings from this study are aligned with research pertaining to significant relationships among perceived ease of use, perceived usefulness, attitude, and intention to use in the context of technology usage [28,38]. In both groups, it was found that perceived ease of use had a significant effect on perceived usefulness and attitude, perceived usefulness improved attitude, and attitude had influence on intention to use. When both faculty and students perceive a LMS to be easy to use, its usefulness, attitudes, and behavioral intentions can be assured to increase.

In the moderating effect according to user group, significant differences were only found in the sense of self-efficacy and perceived ease of use, and not in the link between other external factors. That there was no significant difference despite the expectation that the impact of external variables on LMS acceptance would reveal a distinct difference between professors and students was unexpected. Nevertheless, this discussion is meaningful. Although Piotrowski [47] argued that faculty was indifferent to online tools and that students were positive about academic applications of digital educational tools, these study results showed that faculty acceptance of LMS did not significantly differ from students in the management and reinforcement of external variables. By improving perceived ease of use and usefulness, and forming a favorable attitude, technological behavioral intentions can be promoted.

It is worth noting that this finding came relatively later on in the COVID-19 pandemic. That is, it can be assumed that this is because the adoption of the technology was not voluntarily made, and the LMS was inevitably accepted. Under the irresistible influence of the strong external environment, both groups had no choice in adopting the technology, but an in-depth understanding of the intention to adopt the technology, taking into account external variables, could improve the use of the technology. An integrative view considering user types will contribute to strategies for the adoption of new innovative technologies or blended learning design in the higher education environment.

As Davis [5] asserts, the research result examines the usefulness of LMS after evaluating how easy or difficult it is for users to work with LMS. When a technology is judged to be “easy to use” and “useful”, a positive attitude is formed, and a positive attitude activates the intention to use. In addition, four external determinants directly or indirectly influenced the perceived ease of use, perceived usefulness the attitudes toward the LMS and their intention to use the LMS. This validates the extension of TAM when accounting for student and faculty acceptance of skills to the LMS.

## 7. Implications and Future Research

This study expands on previous observations by taking into account differences in technology acceptance behavior between faculty and students, and it reinforces the application of the extended TAM. The results of the study confirmed the strength of the link between external factors and the original TAM constituents. The findings provide detailed insights into external factors and offer useful and actionable guidance for higher education professionals, including policy makers, to effectively operate LMSs.

Despite the adoption and implementation of LMS in higher education institutions prior to the COVID-19 pandemic, LMS still faced underutilization by faculty and students. On the other hand, in the COVID-19 pandemic context, although LMS has been fully man-dated as an effective tool of emergency remote teaching [28], both faculty and students suffered considerable inconvenience to adapt.

At a standpoint of LMS as a platform for non-face-to-face communication and interaction between faculty and students, considering interactive user types contributes to the existing literature and has practical implications. Furthermore, identification of differences in LMS technology acceptance behaviors according to user types based on the extended TAM model can provide insights and clues about behavioral predictors for technology acceptance in blended learning design [19].

### 7.1. Implications

First and foremost, our findings highlight that, above anything else, universities should build a user-friendly LMS environment in consideration of both users. Next, it is important to pay attention to external factors such as self-efficacy, subjective norm, enjoyment, and computer anxiety, which all affect LMS acceptance by teaching staff and students, and thoroughly evaluate them. It is revealed that self-efficacy and enjoyment were salient determinant of users’ acceptance of LMS. Therefore, universities need to set up their environment with confidence that both users (i.e., faculty and students) can handle skillfully using their own usage skills without the help or experience of their surroundings. Regular training or manuals can enhance users’ sense of self-efficacy [3]. This training should be included incrementally and at intervals in order to continually encourage the perception of ease and usefulness of a LMS; this would improve users’ positive attitudes and intention to use. LMS administrators must incorporate the element of “enjoyment” so that faculty and students alike can interact with each other in an enjoyable, imaginative, fun, and pleasant way.

Overall, this research contributes to TAM literature by providing a developed model and confirming the impacts of its external factors on the utilization of LMS by considering both faculty and students. The empirical results of this study can inform stakeholders, enabling them to make more effective policy decisions related to LMS acceptance.

### 7.2. Limitation and Future Studies

Despite the original and useful findings, some limitations of this study should be acknowledged. The data were obtained from professors and students from universities in Busan (i.e., just one city), so it is unknown to what degree the study reflects purely local characteristics. In addition, our approach was cross-sectional, and captured the user’s perception and intention of use at just a single point in time, meaning the results may vary over time if more time points were included. This study used the most widely used external factors; no other system variables such as technical support or educational design, corresponding to system characteristics were reflected. Future research should look at other external factors and test variables that have not been previously tested but that are expected to be effective.

## Figures and Tables

**Figure 1 ijerph-18-08570-f001:**
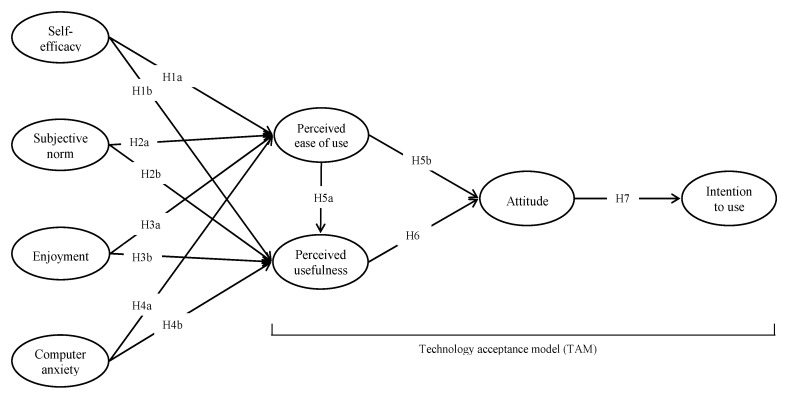
Proposed conceptual model. Note: proposal is for two identical models (i.e., models for faculty and student groups).

**Table 1 ijerph-18-08570-t001:** The comparison of the two groups (MGA).

Path		Student Group		Faculty Group		Significance of the Difference	
		Beta	t-Value	Beta	t-Value	∆Beta	*p*-Value
H1a	Self-efficacy → Perceived ease of use	0.490	9.487 ***	0.259	3.451 ***	0.232	0.011 **
H1b	Self-efficacy → Perceived usefulness	−0.138	2.018 **	−0.179	2.091 **	0.041	0.709
H2a	Subjective norm → Perceived ease of use	0.049	0.802	0.128	2.247 **	−0.079	0.339
H2b	Subjective norm → Perceived usefulness	0.162	1.801	0.219	2.961 ***	−0.057	0.628
H3a	Enjoyment → Perceived ease of use	0.214	3.558 ***	0.206	3.428 ***	0.008	0.923
H3b	Enjoyment → Perceived usefulness	0.585	7.396 ***	0.517	4.656 ***	0.068	0.621
H4a	Computer anxiety → Perceived ease of use	0.275	4.668 ***	0.458	5.676 ***	−0.184	0.067 *
H4b	Computer anxiety → Perceived usefulness	0.021	0.312	−0.026	0.256	0.048	0.693
H5a	Perceived ease of use → Perceived usefulness	0.277	2.941 ***	0.320	2.693 ***	−0.043	0.779
H5b	Perceived ease of use → Attitude	0.232	3.422 ***	0.345	4.632 ***	−0.113	0.262
H6	Perceived usefulness → Attitude	0.697	12.252 ***	0.565	7.640 ***	0.132	0.156
H7	Attitude → Intention to use	0.846	29.155 ***	0.863	38.520 ***	−0.017	0.650
		R^2^	Q^2^	R^2^	Q^2^		
	Perceived ease of use	0.706	0.609	0.746	0.631		
	Perceived usefulness	0.707	0.586	0.620	0.510		
	Attitude	0.730	0.624	0.666	0.600		
	Intention to use	0.716	0.680	0.746	0.704		

Notes: *** *p* < 0.01, ** *p* < 0.05, * *p* < 0.10.

## Data Availability

The data will be made available on request.

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
