# Peer review of "A Comparison of Faculty and Student Acceptance Behavior toward Learning Management Systems"

_ijerph, 2021, doi:10.3390/ijerph18168570_

Round 1

Reviewer 1 Report

I enjoyed reading your paper.  It is overall very good.  However, the introduction has some issues.  Terms are used before they are defined and the flow is difficult.  I have marked up the paper with some of these concerns and hope that you can address them.  

The one other area for improvement is the implicatons for practice and research.  The implications are not built on the research.  Or if it is, it is certainly not clear on how they are related.  Editing this section is in order. 

Reviewer 2 Report

This is an interesting paper which examines teacher and student acceptance of a Learning Management System during the first year of the Covid-19 pandemic, when the much greater use of such systems was mandated across many education contexts. I think the paper is of an acceptable standard and I don’t have any major issues with the approach or findings. I do think that some aspects of the argument could be strengthened, however, and I recommend that the author be given the chance to address the following issues before publication:

  • I think the Abstract could be strengthened by being more specific about the “existing knowledge” being referred to (existing knowledge on what topic, specifically?) and in either explaining or rephrasing the phrase “to be”.
  • I think the Introduction could be strengthened by explaining earlier, and more strongly, why user type is such an important issue to study: why are teacher and student acceptance such important issues to study in the context of Covid-19?
  • Rather than simply stating around line 69-70, that there are not enough studies in the context of Covid-19, it would be useful to state the core motivation for this study. What are you hoping that this study can achieve? Why do you think it can produce knowledge that is worthwhile?
  • Also in the Introduction, I think it would be useful to clarify the distinctions between the similar but slightly different terms that are used: acceptance behaviour, adoption behaviour, behavioural intentions, user attitude, etc. What do these all mean, to what extent are they different or similar, and which are you focussing on in this study and why? (Are they all necessary to use or could some be removed?)
  • In the Introduction and Literature Review, it would be useful to reflect on whether there are any other bodies of knowledge that might have informed this study. Perhaps there are studies abut educational change and technology during the Covid-19 pandemic that do not use the TAM model, for example. Might these not still form part of the backdrop informing the paper?
  • Section 2.2 could be strengthened by discussing why you chose to use TAM for this study. In particular, it would be useful to emphasise how these reasons are related to your research objectives and motivations. You might also discuss the obvious issue that, since LMS use in these Covid-19 contexts is mandated, then the issue of user acceptance as a predictor of intention to use might be less relevant; I am guessing that you do not agree with this view, but you could explain why not.
  • I also think that the issue of user type could be addressed in the Literature Review; doing so might involve moving some of the later content (in section 3.3) earlier.
  • Under the Methodology we need to read more about the research site. Tell the reader something about the university, the type of courses it runs and students it attracts, how it responded to the Covid-19 pandemic at a policy level, and something about the LMS whose use was mandated. I’d say this will require at least two paragraphs. Without knowing this information it is difficult to contextualise the information that follows.
  • When describing the instrument you use, can you state whether the items were used exactly as in their source studies, or whether any translation or rephrasing was required? In particular, was the instrument delivered in Korean or English?
  • I think the results section is largely okay, but I would like to hear more about why certain courses of actions were taken. One example if at line 341. What were you trying to achieve? And so why did you choose those criteria and the SmartPLS tool? In other words, say ‘why’ you did it as well as ‘what’ you did.
  • I think the Discussion could be strengthened by emphasising the differences with other literature earlier. I also think it would be useful to take into account the issues I raised in my bullet points about the Introduction. To what extent has examining user types been useful and might it influence future literature? To what extent is using TAM useful in the context of mandated change during circumstances like this Covid-19 pandemic?

Reviewer 3 Report

The theme is current and relevant. However, there are a number of issues that prevent its publication. These are described below:

-1 The authors talk about learning management systems (LMS) or the technology acceptance model (TAM) without first referring to them.

-2 - The review of the scientific literature on learning management systems is not relevant. The paragraphs of the Introduction do not lead to a logical sequence of interrelated ideas that would allow the reader to know what the "knowledge gaps" are on the main subject of the paper.

-3-There are ideas that keep repeating themselves and are very vague.

-4 - In section two, "Literature review" is where the central terms of the manuscript that have been used in the previous section are defined. The use of abbreviations is so that they can be used from the first time they are cited in the text. The authors do not follow any criteria in many of them (e.g. LMS or LMSs).

-5-The mention of COVID-19 is not the appropriate terminology to refer to the global crisis we are experiencing worldwide, as it is not the specific name of the virus, nor do the authors refer to transmission or contagion by the virus. The authors should use a term that correctly defines the situation they are referring to.

-6-The authors do not follow any checklist according to the study design carried out (STROBE, CONSORT etc...). Nor do they indicate the type of study carried out (observational study, case-control study...). This makes the structure of the article difficult to follow, without an Introduction that concludes with the gaps in knowledge, without concrete and achievable objectives.

-7- In the section "Methodology" 4.1. Sampling and data collection, no criteria for inclusion and exclusion of participants were defined. The authors talk about requirements, but what were these requirements?

-8-The authors discuss self-efficacy, enjoyment, and computer anxiety as factors to be taken into account, although at no point do they relate them to fundamental aspects of the work, neither to the objectives nor to the type of statistical analysis carried out.

-9-A statistical analysis section is not shown. Similarly, section 5.2. Measurement model analysis: measurement invariance test should not appear in the Results section.

10-The results are not self-explanatory, not showing the relevance of the results.

-11. The discussion does not go into the results achieved, nor does it interpret them correctly on the basis of previous studies. Any discussion should end with a section on the conclusions of the study, which does not appear in this work. The section on limitations of the study and implications presents ideas that are not specifically drawn from this study.

Round 2

Reviewer 3 Report

The theme is current and relevant. However, there are a number of issues that prevent its publication. These are described below:

Response: We thank you for your comments and suggestions on the previous version of this manuscript. We have thoroughly studied them and have revised the manuscript accordingly. This report summarizes our responses to all the comments (in red for your convenience).

Re: Thanks. I appreciate the efforts of the authors in submitting an improved version of their study, but even so, the version submitted is insufficient for it to be published in this form.

-1 The authors talk about learning management systems (LMS) or the technology acceptance model (TAM) without first referring to them.

Response: Thank you for this critical comment. We included the short description in Introduction.

☞ Various online tools were utilized to try and maintain the same sense of depth of interaction between faculty and students as in face-to-face learning, and most universities made heavy use of a learning management system (LMS), which is an online software for academic lecturing and learning, as a solution to manage educational programs [1,2].

Re: Thanks

A considerable number of studies on the technology acceptance model (TAM) have confirmed its’ predictive power in technology adoption behavior in individuals in various sectors, including education [3,4]. The TAM holds that two essential determinants (i.e. perceived ease of use and perceived usefulness) of a user’s attitude, which is an individual’s a favorable or unfavorable evaluation toward a particular technology induce behavioral intentions, which is the likelihood of a person’s acceptance of a technology, and subsequent actual use [5].

Re: The authors indicate "a considerable number of studies..." citing only two at a later date. Please be more specific in the way you write.

-2 - The review of the scientific literature on learning management systems is not relevant. The paragraphs of the Introduction do not lead to a logical sequence of interrelated ideas that would allow the reader to know what the "knowledge gaps" are on the main subject of the paper.

Response: Thank you for allowing us to improve Introduction. We reshaped Introduction by stating current situation in higher education context à the application of extended TAM in education context à articulating the importance of the comparative approach in examine the adoption of two primary users of a LMS à explaining the research gap and study objectives. Specifically, the knowledge gaps are stated as below.

☞ Existing studies pertaining to the technology acceptance behavior in higher education are mostly posited either from the standpoint of only the faculty [3,4] or focused on just the student’s perspective [7]. There are only a handful of studies [16,17] that investigate the difference between faculty and students regarding the development of behavioral intent toward technology-powered services for lecturing/learning purposes. Meanwhile, in the wake of COVID-19, there is a large body of study that addresses the future outlook of education in conjunction with the technical solutions [18,19]. However, there are no sufficient empirical studies specifically during the critical first year of COVID-19, when the use of a LMS as lecturing/learning online tool became more or less mandated overnight.

-3-There are ideas that keep repeating themselves and are very vague.

Response: Thank you for this critical comment.

We removed sentences which are repeatedly stated. Examples are as following.

☞ In addition, a clearer understanding of the differences between faculty and students would help in the effective inclusion of innovative technologies within higher education environments.

The relative low level of utilization of a LMS for actual lecturing and learning purposes has already been discussed along with users’ different levels of acceptance behavior in a few studies.

Re: thank you for adjusting

-4 - In section two, "Literature review" is where the central terms of the manuscript that have been used in the previous section are defined. The use of abbreviations is so that they can be used from the first time they are cited in the text. The authors do not follow any criteria in many of them (e.g. LMS or LMSs).

Response: Thank you for pointing this out. We have updated the use of abbreviations to be consistent throughout the manuscript.

Re: Thanks

-5-The mention of COVID-19 is not the appropriate terminology to refer to the global crisis we are experiencing worldwide, as it is not the specific name of the virus, nor do the authors refer to transmission or contagion by the virus. The authors should use a term that correctly defines the situation they are referring to.

Response: We are sorry, but we do not understand the meaning of your words. We referred the official sources such as WHO or CDC to state “the COVID-19 crisis” and explain the current evidence. We would appreciate if you could elaborate a bit.

-6-The authors do not follow any checklist according to the study design carried out (STROBE, CONSORT etc...). Nor do they indicate the type of study carried out (observational study, case-control study...). This makes the structure of the article difficult to follow, without an Introduction that concludes with the gaps in knowledge, without concrete and achievable objectives.

Response: Thank you for pointing this out. Our study is neither observational nor experimental study, and thus this is not required to follow STROBE, CONSORT, etc. In order to test hypotheses we have developed, we conducted quantitative approach based on a survey data. We included the short description study design in 4.1. Sampling and data collection.

Re: the quality of answers given to the questions are insufficient. Please make this effort in respect to your work and efforts. All studies are defined on the basis of the type of study conducted. Please consult a statistician if in doubt.

-7- In the section "Methodology" 4.1. Sampling and data collection, no criteria for inclusion and exclusion of participants were defined. The authors talk about requirements, but what were these requirements?

Response: Thank you for this critical comment. We included the short description of requirements.

☞ Of the collected surveys, 24 (14 for students and 10 for faculty) were rejected due to irrelevant contents which are not lecturing and learning in higher education contexts or demographic missing information.

Re: Who decided on the rejection of the survey - an evaluator from outside the study? On the basis of what criteria?. This lack of clarity could be seen as a risk of bias.

-8-The authors discuss self-efficacy, enjoyment, and computer anxiety as factors to be taken into account, although at no point do they relate them to fundamental aspects of the work, neither to the objectives nor to the type of statistical analysis carried out.

Response: Thank you for pointing this out. In previous studies, self-efficacy, enjoyment, and computer anxiety were emphasized as important variables that were grafted onto TAM. Therefore, we verified the effect of these variables on perceived ease of use and perceived usefulness through previously used measurement items.

-9-A statistical analysis section is not shown. Similarly, section 5.2. Measurement model analysis: measurement invariance test should not appear in the Results section.

Response: Thank you for allowing us to improve a statistical analysis section. Section 5.2. Measurement model analysis: measurement invariance test has been changed to Results of statistical analyses. The results of statistical analysis are showed in this result section.

10-The results are not self-explanatory, not showing the relevance of the results.

Response: Thank you for this critical comment. We improved the description of results.

☞ As displayed in Table 5, significant group differences were only found for the impact of self-efficacy use (estimate of difference = .232, p-values of difference = .011, p < .05) and computer anxiety use (estimate of difference = -.184, p-values of difference = .067, p < .10) on perceived ease of. Therefore, H8a and H8g were supported. However, user type does not moderate other relationships in the student and faculty groups. Thus, H8 was found to have a partial moderating effect.

Re: Thanks for clarifying, however, the authors should consider reducing the number of hypotheses, integrating some of them into others in order to simplify the study and make it easier for the less expert reader to understand.

-11. The discussion does not go into the results achieved, nor does it interpret them correctly on the basis of previous studies. Any discussion should end with a section on the conclusions of the study, which does not appear in this work. The section on limitations of the study and implications presents ideas that are not specifically drawn from this study.

Response: Thanks for your comments for improvement. We revised the discussion in general based on the results achieved. The Implications section was also supplemented based on the ideas derived from the research, and less relevant parts were deleted.

☞ A LMS is a key element in the delivery of education at tertiary institutions, and its use—and lack of use—has been brought to the fore during the COVID-19 pandemic.

☞ Universities should pay rigorous attention to the infrastructure of a LMS and check the readiness of both instructors and students for technology acceptance in all processes.

☞ A LMS administrator should build user-friendly interfaces that also reflect an element of “enjoyment” (e.g. gamification, audiovisual assistance, animation simulation, and experimental videos)

☞ Curricula should be organized to encourage student and faculty awareness of perceived ease of use and usefulness so that students’ positive attitudes and behaviors can be im-proved. Given the importance of external factors for LMS acceptance, a culture of online learning should be instilled in both faculty and students. In addition, if an institutional reward system is provided to strengthen subjective norms for faculty, technology ac-ceptance can be further strengthened.

☞ We addressed the limitation of considering only one standpoint of faculty and students and provide evidence of the user type’s moderating role.

In case of students group, the results indicated that self-efficacy, enjoyment, and computer anxiety have a significant influence on perceived ease of use of a LMS. The variables that had a significant effect on perceived usefulness were self-efficacy and enjoyment. However, in faculty group, self-efficacy, subjective norm, enjoyment, and computer anxiety were found to have an impact on perceived ease of use of a LMS. Besides self-efficacy, subjective norm, and enjoyment significantly affected perceived usefulness.

☞ In fact, it is widely accepted that it is easy for students and professors with high self-efficacy to use a LMS [3,4,24,29]. Consequently, these confident individuals use a LMS more than less-confident individuals [3]. This is supported by Bandura [26] in that high self-efficacy leads to an active learning process.

☞ In addition, subjective norms, which were significant determinants only for faculty group, disprove the importance of social influence on them. In other words, they want to meet the needs, expectations, and preferences of members of society who are important or valuable to them. So we can expand the acceptance of LMS by strengthening it. Meanwhile from student group it also supports the findings of Salloum et al. [38], which assume that they cannot be confident that a LMS would be easy to use and useful even if a LMS is socially preferred according to peers. While these findings were not in line with the results of other prior studies [8,28] in which subjective norm was shown to have a positive impact on perceived ease of use and usefulness in the student group.

☞ In other words, both faculty and students groups feel the ease of use when they do not feel scared, uncomfortable, or nervous when using the computer, but they do not consider that it is useful to improve performance and achievement, and increase the learning effect.

☞ In both groups, it was found that perceived ease of use had a significant effect on perceived usefulness and attitude, perceived usefulness improved attitude, and attitude had influence on intention to use. When both faculty and students perceive an LMS to be easy to use, its usefulness, attitudes, and behavioral intentions can be assured to increase.

☞ Nevertheless, this discussion is meaningful. Although Piotrowsk and Pedagogical [47] argued that faculty were indifferent to online tools and that students were positive about academic applications of digital educational tools, their study results showed that faculty acceptance of LMS did not significantly differ from students in the management and reinforcement of external variables. By improving perceived ease of use and usefulness, and forming a favorable attitude, technological behavioral intentions can be promoted.

Re: “Although Piotrowsk and Pedagogical…”. correct the error

☞ Under the irresistible influence of the strong external environment, both groups had no choice in adopting the technology, but an in-depth understanding of the intention to adopt the technology, taking into account external variables, could improve the use of the technology. An integrative view considering user types will contribute to strategies for the adoption of new innovative technologies or blended learning design in the higher education environment.

☞ First and foremost, our findings highlight that above anything else, universities should build a user-friendly LMS environment in consideration of both users. Next, it is important to pay attention to external factors such as self-efficacy, subjective norm, enjoyment, and computer anxiety, which all affect LMS acceptance by teaching staff and students, and thoroughly evaluate them. It is revealed that self-efficacy and enjoyment were salient determinant of users’ acceptance of LMS. Therefore, universities need to set up their environment with confidence that both users (i.e. faculty and students) can handle skillfully using their own usage skills without the help or experience of their surroundings. Regular training or manuals can enhance users' sense of self-efficacy [3]. This training should be included incrementally and at intervals in order to continually encourage the perception of ease and usefulness of a LMS; this would improve users’ positive attitudes and intention to use. LMS administrators must incorporate the element of “enjoyment” so that faculty and students alike can interact with each other in a enjoyable, imaginative, fun, and pleasant way.

☞ Despite the original and useful findings, some limitations of this study should be acknowledged. The data were obtained from professors and students from universities in Busan (i.e., just one city), so it is unknown to what degree the study reflects purely local characteristics. In addition, our approach was cross-sectional, and captured the user’s perception and intention of use at just a single point in time, meaning the results may vary over time if more time points were included. This study used the most widely used external factors; no other system variables such as technical support or educational design, corresponding to system characteristics were reflected. Future research should look at other external factors and test variables that have not been previously tested but that are expected to be effective.

We truly appreciate your time and help. We hope that we successfully addressed your comments and concerns. Thank you very much!
